# Arrhythmias after SARS-CoV-2 Vaccination in Patients with a Cardiac Implantable Electronic Device: A Multicenter Study

**DOI:** 10.3390/biomedicines10112838

**Published:** 2022-11-07

**Authors:** Naruepat Sangpornsuk, Voravut Rungpradubvong, Nithi Tokavanich, Sathapana Srisomwong, Teetouch Ananwattanasuk, Padoemwut Teerawongsakul, Stephen J. Kerr, Mathurin Suwanwalaikorn, Krit Jongnarangsin, Ronpichai Chokesuwattanaskul

**Affiliations:** 1Center of Excellence in Arrhythmia Research, Cardiac Center, King Chulalongkorn Memorial Hospital, The Thai Red Cross Society, Bangkok 10330, Thailand; 2Division of Cardiovascular Medicine, Faculty of Medicine, Chulalongkorn University, King Chulalongkorn Memorial Hospital, The Thai Red Cross Society, Bangkok 10330, Thailand; 3Cardiology Division, Department of Internal Medicine, Faculty of Medicine Vajira Hospital, Navamindradhiraj University, Bangkok 10300, Thailand; 4Biostatistics Excellence Centre, Faculty of Medicine, Chulalongkorn University, Bangkok 10330, Thailand; 5Frankel Cardiovascular Center, Division of Cardiovascular Medicine, University of Michigan Health, Ann Arbor, MI 48109, USA

**Keywords:** arrhythmia, CIED, COVID-19, SARS-CoV-2, vaccination

## Abstract

One of the most concerning adverse events related to the SARS-CoV-2 vaccination is arrhythmia. To ascertain the relationship between vaccination and arrhythmic events, we studied the occurrence of arrhythmia in patients with cardiac implantable electronic devices (CIEDs) before and after a SARS-CoV-2 vaccination. Patients with CIEDs aged ≥18 who visited the CIED clinic at King Chulalongkorn Memorial Hospital and Vajira hospital from August 2021 to February 2022 were included. Information about the SARS-CoV-2 vaccination and side effects was obtained. One hundred eighty patients were included in our study, which compared the incidence of arrhythmias in the 14 days post-vaccination to the 14 days before vaccination. The median age was 70 years. The number of patients who received one, two, and three doses of the vaccine was 180, 88, and 4, respectively. ChAdOx1 was the primary vaccine used in our institutes, accounting for 86% of vaccinations. The vaccination was significantly associated with a 73% increase incidence of supraventricular tachycardia (SVT). In an adjusted model, factors associated with the incidence of SVT were the post-vaccination period, female sex, increasing BMI, chronic kidney disease, and a history of atrial fibrillation. Increased ventricular arrhythmia (VA) episodes were also associated with the post-vaccination period, female sex, decreasing BMI, and chronic kidney disease, but to a lesser degree than those with SVT episodes. No life-threatening arrhythmia was noted in this study. In conclusion, the incidence of arrhythmia in patients implanted with CIEDs was significantly increased after the SARS-CoV-2 vaccination.

## 1. Introduction

Coronavirus 2019 is a positive-strand RNA virus that has caused a worldwide pandemic. Infected patients have various clinical manifestations ranging from common cold-like symptoms to acute respiratory distress syndrome (ARDS) [1,2]. Effective treatments for this virus are limited, and preventing a severe infection by vaccination remains a critical public health intervention to confine the pandemic. As the delta variant spread through Thailand, the available primary vaccines were the viral vectored ChAdOXI-nCoV-19 (AstraZeneca, Cambridge, UK) and an inactivated vaccine (Sinovac (Biotech, Hyderabad, India)). Later in the pandemic, mRNA was also available (BioNTech, Mainz, Germany (Pfizer, New York, NY, USA) and Moderna, Cambridge, UK). Many studies have documented the adverse events following the administration of the SARS-CoV-2 vaccine [3]. Local adverse effects have been reported in 58.1–71.9% of patients, including pain, swelling, and redness [4,5]. Systemic side effects have also been reported with varying incidences across studies (13.5–33.7%), including fatigue, headache, myalgia, fever, and diarrhea [4,5]. A common adverse effect is a palpitation [6]. A study conducted on 5866 health care workers (5589 who received ChAdOx1 (AstraZeneca) and 277 who received BNT162b2 (Pfizer/BioNTech)) revealed that 27.3% of participants reported palpitations (28.3% in the ChAdOx1 group and 4.3% in the BNT162b2 group) [7]. However, palpitations could be a subjective feeling related to anxiety about the procedure, and objective evidence on the prevalence of the SARS-CoV-2 vaccine-related arrhythmia is still lacking. Therefore, we decided to study the incidence of arrhythmias in patients with cardiac implantable electronic devices (CIEDs) with a recent SARS-CoV-2 vaccination. 

## 2. Materials and Methods

### 2.1. Trial Design

The study is conducted with multicenter sites.

### 2.2. Participants 

Consecutive patients with CIEDs who visited the pacemaker clinic at King Chulalongkorn Memorial Hospital (Bangkok, Thailand) or Vajira hospital (Bangkok, Thailand), aged >18 years were included in the study. Two hundred patients were screened, and one hundred eighty-eight consented to participate in our research. Unfortunately, twelve patients declined as it was inconvenient to be interviewed with the questionnaire. The Institutional Review Board (IRB) of Chulalongkorn University and Navamindradhiraj University approved the study (IRB number COA 192/64, approval date 30 September 2021), and all participants provided written informed consent.

### 2.3. Methods

Patients who consented to participate in our study were interviewed using questionnaires about the SARS-CoV-2 vaccination. Questions included vaccine type, number of vaccinations, date of vaccination, and adverse effects. In addition, electrical medical records were reviewed to obtain demographic data and medical histories. Data about arrhythmic episodes was retrieved by device interrogation and was uploaded to our database system. Episodes of any arrhythmias within 14 days before and after vaccination were recorded and verified by a cardiologist. 

### 2.4. Definition of Terms

Ventricular arrhythmias (VA) included non-sustained and sustained ventricular tachycardia and ventricular fibrillation episodes. In single-chamber pacemakers, high ventricular rate episodes (HVR) were recorded as ventricular arrhythmias. HVR tracings were reviewed to confirm the diagnosis of ventricular arrhythmias, and differentiate from supraventricular tachycardia (SVT) with a rapid ventricular response. “Sustained ventricular arrhythmia (VA) are VA episodes that require device therapies, either antitachycardia pacing (ATP) or AICD shock, to terminate VA.” Different companies use the same definition for a sustained VA. VF episodes are all initially treated by ICD shock, as all CIEDs have been programmed, according to the HRS guidelines” [8,9]. 

Supraventricular tachycardia (SVT) included any atrial high-rate episodes (excluding noise) and SVT episodes (recorded by the automated implantable cardioverter defibrillator, AICD). Definite PMT episodes were excluded from the study. 

Chronic kidney disease was defined as eGFR < 60 mL/min/1.73 m^2^.

Valvular heart disease was defined as severe regurgitation or stenosis of any valves or post valvular intervention or surgery. Myocarditis was an exclusion criterion of the study. We defined myocarditis, based on the clinical presentation, 12 lead electrocardiography, and cardiac biomarker criteria. 

### 2.5. Statistical Analysis

A previously published study reported the incidence of palpitations, and possible arrhythmia, in patients receiving the SARS-CoV-2 vaccine by 27% [7]. We had approximately 200 eligible patients; assuming 5% of participants would have arrhythmia in the pre-vaccination period, and the proportion who would experience arrhythmia increased by 10% post-vaccination, 155 participants would provide 80% power to detect this difference at a two-sided significance level of 5%. This was inflated to 190 participants in the event of a participant drop out. Participant continuous demographic and disease-related data were summarized as mean (standard deviation [SD]) or median (interquartile range [IQR]), as appropriate. Categorical data were summarized as frequency (%). Comparisons between the number of SVT and VA, experienced by each participant in the pre- and post-vaccination periods were assessed using a Poisson generalized estimating equation (GEE) to give the incidence rate ratios (IRRs) and adjusted IRR (aIRR). Potential confounders assessed for their association were age, BMI, period (pre- or post-vaccination), comorbidities, whether the participant experienced any vaccine-related AE, and any indication for cardiac implantable electronic devices; indication was not modeled for the SVT outcome because, from a clinical perspective, there is no correlation between SVT and indication for CIEDs when compared with VA. Potential confounders significant at *p* < 0.1 in the univariable models were adjusted for in a multivariable model.

## 3. Results

From August 2021 to February 2022, a total of 188 patients received at least one dose of the SARS-CoV-2 vaccine and eight patients were excluded, due to incomplete follow up data. Baseline (pre-first vaccination) characteristics of these patients are shown in Table 1. The median age of study participants was 70 (IQR 59–79) years, and the mean BMI was 23.6 (SD 4.32) kg/m^2^. The number of participants who received one, two and three doses of the vaccine was 180, 83 and, 4 participants, respectively. Indication for CIED implantation was an AV node dysfunction (32%), sinus node dysfunction (30%), primary prevention for VT/VF (18%), secondary prevention for VT/VF (11%) and cardiac resynchronization (6%). There were three types of vaccine in our study which ChAdOx1 was a mainly used vaccine, and accounted for 86%. Comorbidities are also shown in Table 1 and were atrial fibrillation (31%), ischemic heart disease (24%), chronic kidney disease (17%), valvular heart disease (11%) and Brugada syndrome (7%). There is a greater proportion of males, compared to females, with LVEF <40% (51% vs. 19%, respectively).

Adverse effects of the vaccination were reported by 38 participants (21%) with no difference between the types of vaccinations. The most common side effect was fever, found in 13 patients (7%), followed by a local side effect in nine patients (5%). Other side effects are shown in Table 2. 

Fourteen participants had at least one episode of SVT before vaccination. Following the vaccination, the number of participants who had at least one episode of SVT increased to 21 and overall episodes of SVT increased from 86 to 147 episodes. For VT, six participants had a VA episode before vaccination, and after vaccination the number of participants experiencing any VA increased to 10; the overall episodes of VT increased from 12 to 50 episodes. Details about the number of patients experiencing arrythmia, and number of arrhythmias before and after each dose of the vaccination is shown in Table 3.

Factors associated with the incidence of SVT in univariable and multivariable analyses are shown in Table 4. Once the factors meeting the inclusion criteria for our multivariate model are adjusted, the incidence rate of SVT post vaccination significantly increased by 73%, compared to the pre-vaccination period (adjusted incidence rate ratio (aIRR) 1.73, 95% CI 1.34–2.22). The factors associated with a significant increase in SVT, included the female sex (aIRR 9.72, 95% CI 5.84–16.17), CKD (aIRR 4.40, 95% CI 3.14–6.16) and AF (aIRR 3.35, 95% CI 2.42–4.64). An increasing BMI was associated with a reduction in the incidence of SVT (aIRR 0.90, 95% CI 0.87–0.94).

For VA, after adjusting for whether the participant experienced a vaccine-related AE, the comorbidities, sex, age, BMI and indication for a CIED in a multivariable model, the adjusted incidence rate ratios increased 4.16 fold in the post-vaccination period (aIRR 4.16, 95% CI 2.22–7.79) (Table 5, Figure 1). The factors associated with a significant increase in VT included being overweight (aIRR 1.3 per a BMI increase, 95% CI 1.23–1.36), and at an older age (aIRR 1.09 per year increase, 95% CI 1.05–1.13). The female gender (aIRR 0.30, 95% CI 0.14–0.64), CKD (aIRR 0.04, 95% CI 0.01–0.33) and AF (aIRR 0.36, 95% CI 0.14–0.93), were associated with a reduction in the incidence of VT.

## 4. Discussion

In our study primarily focusing on CIED patients receiving the SARS-CoV-2 vaccination, the incidences of arrhythmias, both SVT and VA, significantly increased in the occurrence after the SARS-CoV-2 vaccination. The overall adverse events were 21%, which is less than other studies [5,7], and the rates of systemic adverse events (mainly fever) were higher than our rates of local reactions. The probable reason for this discrepancy between our study and others is the recall bias since most patients will recall systemic side effects more than local ones. Our data showed interesting findings of discordance in SVT and VT risk factors with some patients’ characteristics. Females and CKD increased SVT but lowered VT incidences. Repolarization heterogeneity has already been established in females, so QT variability, a common VA trigger provoked by the vaccination, might not affect the incidence of VT [9]. In our cohort, females had a lower prevalence of the left ventricular systolic dysfunction, compared to males. Therefore, the better left ventricular function might contribute to the lower VT incidence. Confounding factors in CKD patients includes intensive risk factor modification therapy, medications, and more frequent follow-up intervals, which might play a role in lowering the VT risk. However, the relationship between CKD and a lowered VT incidence in our study can only be an assumption as there is no existing evidence to support this finding. 

Data from the U.S. FDA Center for Biologics Evaluation and Research reveal many potential cardiovascular system adverse events after a SARS-CoV-2 vaccination [10]. These events include acute myocardial infarction, deep vein thrombosis, non-hemorrhagic stroke, myocarditis/pericarditis, and pulmonary embolism. Data from a systematic review showed that myocarditis and myopericarditis events were more frequently reported after the mRNA vaccines, while MI and ischemic heart disease were more commonly reported, following the ChAdOXI-nCoV-19 (AstraZeneca) vaccination [10]. It has been hypothesized that myopericarditis relates to the lipophilic medium, which comprises polyethylene glycol (PEG) used to deliver the active ingredients intracellularly to elicit an immune response [11]. People with an allergy to PEG may develop myocarditis secondary to the allergic reaction. This response may also explain the lower prevalence of myocarditis post ChAdOXI-nCoV-19 and inactivated vaccines as they are devoid of PEG. Another possible mechanism relates to the spike (S) protein of SARS-CoV-2, which binds with high affinity to the angiotensin-converting enzyme 2 (ACE2), which changes angiotensin (AT) II into angiotensin [12]. ATII has a significant role in modulating inflammation, so the SARS-CoV-2 vaccination can potentially cause abnormal inflammation and lead to myocarditis. The mechanism behind myocardial infarction following the ChAdOx1vaccination has been established, due to a prothrombotic state. Greinacher et al., demonstrated that the clinical picture of thrombocytopenia and thrombotic complications, which develop approximately one to two weeks after the vaccination with ChAdOx1 nCov-19, is a disorder similar to a severe heparin-induced thrombocytopenia [13]. Our study found no cases of significant myocarditis in participants. The possible pathophysiological mechanisms of the vaccine related arrhythmia include immune dysregulation, myocarditis, and systemic inflammation [4]. 

Data regarding arrhythmic complications after the vaccination are scarce. One retrospective study from the English National Immunization (NIMS) Database recorded data about arrhythmia incidences in over 38 million patients [14]. The arrythmias were categorized as ventricular tachycardia, ventricular fibrillation, atrial fibrillation and flutter, atrioventricular block or SVT, and the incidences were compared between 28 days pre- and post-vaccination. The results showed that in the period from 1–28 days post vaccination, a decreased risk of cardiac arrhythmia associated with the first dose of ChAdOx1 (IRR 0.94, 95% CI 0.93, 0.96) and BNT162b2 (IRR 0.89, 95% CI 0.87, 0.90) and following a second dose of ChAdOx1 (IRR 0.95, 95% CI 0.94, 0.96) and BNT162b2 (IRR 0.95, 95% CI 0.93, 0.96). There was an increased risk of cardiac arrhythmia following a second dose of mRNA-1273 (IRR 1.46, 95% CI, 1.08, 1.98) and a SARS-CoV-2 positive test (IRR 5.35, 95% CI 5.21, 5.50). A subgroup analysis showed that over the first 28 days post exposure, there was an increased risk of atrial fibrillation or flutter arrhythmia at 15–21 days, following the first dose of the mRNA-1273 vaccine (IRR 2.06, 95% CI 1.11, 3.82), of ventricular fibrillation at 22–28 days following the second dose of the ChAdOx1 vaccine (IRR 1.35, 95% CI 1.05, 1.74) and other cardiac arrhythmias at one–seven days following the second dose of the mRNA-1273 vaccine (IRR 2.32, 95% CI 1.49, 3.62). There was an increased risk of all cardiac arrhythmia subgroups in the 1–28 days following a SARS-CoV-2 positive test. These results are consistent with our study, with a 73% increase in the incidence rate of SVT and a 316% increase in the incidence ratio of VT. The period up to seven days post-vaccination was the period in which prior studies reported the most side effects [5]. Hence, we believe that two weeks, arbitrarily defined by this study, should be sufficient to cover all acute side effects of the vaccination.

Limitations of our study, due to the study design, were that we could not use data about the burden of PACs and PVCs, which occurred more commonly than more sustained arrhythmias. In addition, the devices could not be programmed to collect premature beats in the predetermined period before and after the vaccination. Another limitation was that some arrhythmia, such as AHRE, which mainly needs an atrial rate of more than 170/min, did not meet the criteria of the defined arrhythmia by the devices. Furthermore, a high ventricular rate that we used in the study from single chamber devices could be a supraventricular arrhythmia, despite extensive review of the electrograms. Furthermore, a possible recall bias, as previously noted, might make data about the side effects of our study differ from other research, since our study had a long period from vaccination to survey (mean duration of 30 days). Nevertheless, arrhythmias captured by the devices provided an objective assessment of our primary study outcome. Our observations on the gender difference in the incidence of arrhythmia are subject to bias from this observational study and do not imply a causal relationship. However, this finding warrants further investigation to confirm the result and elucidate a possible mechanism. Furthermore, the subject in this cohort is patients with multiple cardiovascular comorbidities. Hence, the results of this study cannot be extrapolated into the general healthy population. Lastly, since our study is an observational cohort, it is subject to unobserved confounding, and a causal relationship between vaccine and cardiac arrhythmias cannot be inferred from our study results.

## Figures and Tables

**Figure 1 biomedicines-10-02838-f001:**
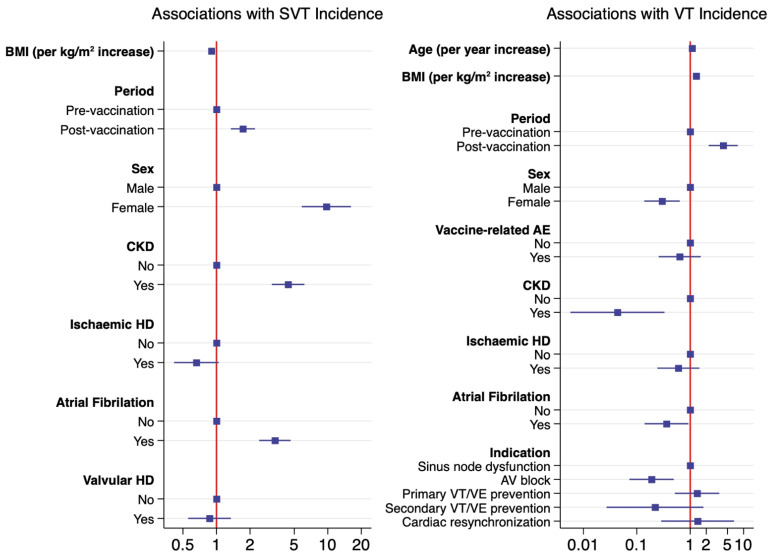
Factors associated with SVT and VA incidence.

**Table 1 biomedicines-10-02838-t001:** Baseline characteristics of the participants.

Characteristic *	Patient Cohort *n* = 180
Median (IQR) age in years	70 (59–79) [min = 24, max = 98]
Male:Female, n(%)	101 (56): 79 (43)
Mean (SD) BMI (kg/m^2^)	23.6 (4.3)
**Number of vaccines received**	
1	180 (100)
2	83 (49)
3	4 (2)
**Indication**	
Sinus node dysfunction	54 (30)
AV block	57 (32)
Primary prevention for VT/VF	33 (18)
Secondary prevention for VT/VF	19 (11)
Cardiac resynchronization	10 (6)
Unknown	7 (4)
**Type of device**	
Single chamber pacemaker	17
Dual chamber pacemaker	93
Single chamber AICD	41
Dual chamber AICD	15
CRT-D	12
CRT-P	2
**Company of device**	
Abbott	51
Boston	68
Medtronic	61
**Vaccine received**	
ChAdOx1	154 (86)
Sinovac/Sinopharm	21 (12)
mRNA	5 (3)
**Comorbidities**	
Atrial fibrillation	44 (31)
Ischemic heart disease	43 (24)
Chronic kidney disease **	30 (17)
Valvular heart disease ***	20 (11)
Brugada syndrome	13 (7)
Left ventricular ejection fraction (LVEF) ****	
≥40%	82
<40%	30

* Category percentages are rounded and may not total 100%. ** CKD = chronic kidney disease defined as eGFR < 60 mL/min/1.73 m^2^. *** Valvular heart disease defined as severe regurgitation or stenosis of any valves or post valvular intervention or surgery. **** LVEF data is missing in 68 patients.

**Table 2 biomedicines-10-02838-t002:** Side effects of the vaccination.

Side Effect *	Overall N = 180
Any side effect (%)	38 (21%)
Fever	13 (7%)
Headache	2 (1%)
Local reaction	9 (5%)
Dyspnea	1 (0.5%)
Insomnia	1 (0.5%)
>1 side effect	12 (7%)

* Category percentages are rounded and may not total 100%.

**Table 3 biomedicines-10-02838-t003:** Distribution of VT and SVT pre- and post- vaccination.

Vaccination 1 (N = 180)		
Timing	Variable	N
Pre-vaccination	Any SVT *	4
	sum of SVT **	7
	Any VT ^#^	3
	sum of VT ^##^	6
Post-vaccination	Any SVT	7
	sum of SVT	63
	Any VT	9
	sum of VT	47
**Vaccination 2 (N = 83)**		
Timing	Variable	N
Pre-vaccination	Any SVT	10
	sum of SVT	79
	Any VT	3
	sum of VT	6
Post-vaccination	Any SVT	13
	sum of SVT	81
	Any VT	1
	sum of VT	3
**Vaccination 3 (N = 4)**		
Timing	Variable	N
Pre-vaccination	Any SVT	0
	N of SVT	0
	Any VT	0
	N of VT	0
Post-vaccination	Any SVT	1
	N of SVT	3
	Any VT	0
	N of VT	0
**All vaccinations (N = 267)**		
Timing	Variable	N
Pre-vaccination	Any SVT	14
	N of SVT	86
	Any VT	6
	N of VT	12
Post-vaccination	Any SVT	21
	N of SVT	147
	Any VT	10
	N of VT	50

* Any SVT = number of participants who had at least one episode of SVT. ** sum of SVT = sum of episodes of SVT in all participants. ^#^ Any VT = number of participants who had at least one episode of VT. ^##^ sum of VT = sum of episodes of VT in all participants.

**Table 4 biomedicines-10-02838-t004:** Univariable and multivariable incidence rate ratios for SVT. (aIRR; adjusted incidence rate ratio. IRR; incidence rate ratio).

	Univariable		Multivariable	
Characteristic	IRR (95% CI)	*p*	aIRR (95% CI)	*p*
Post- vs pre-vaccination	1.72 (1.33–2.21)	<0.001	1.73 (1.34–2.22)	<0.001
Female vs. male sex	10.88 (6.74–17.55)	<0.001	9.72 (5.84–16.17)	<0.001
Age (per year increase)	1.00 (0.99–1.004)	0.30		
BMI (per kg/m^2^ increase)	0.87 (0.83–0.9)	<0.001	0.90 (0.87–0.94)	<0.001
Experienced vaccine-related AE	1.01 (0.71–1.43)	0.97		
CKD	4.26 (3.14–5.8)	<0.001	4.40 (3.14–6.16)	<0.001
IHD	0.57 (0.38–0.84)	0.005	0.66 (0.42–1.05)	0.08
AF	3.88 (2.88–5.22)	<0.001	3.35 (2.42–4.64)	<0.001
Valvular HD	2.89 (2.05–4.08)	<0.001	0.87 (0.56–1.35)	0.87
Brugada syndrome	0.57 (0.28–1.13)	0.11		

**Table 5 biomedicines-10-02838-t005:** Univariable and multivariable incidence rate ratios for VT.

Characteristic	Univariable	Multivariable
	IRR (95% CI)	*p*	aIRR (95% CI)	*p*
Post- vs. pre-vaccination	4.09 (2.21–7.56)	<0.001	4.16 (2.22–7.79)	<0.001
Female vs. male sex	0.35 (0.19–0.64)	0.001	0.3 (0.14–0.64)	0.002
Age (per year increase)	1.02 (0.999–1.04)	0.07	1.09 (1.05–1.13)	<0.001
BMI (per kg/m^2^ increase)	1.35 (1.29–1.41)	<0.001	1.3 (1.23–1.36)	<0.001
Experienced vaccine-related AE	0.45 (0.20–1.01)	0.053	0.64 (0.26–1.58)	0.33
Hypertension	6.25 (3.05–12.81)	<0.001		
DM	0.33 (0.14–0.78)	0.01		
CKD	0.08 (0.01–0.6)	0.01	0.04 (0.01–0.33)	0.002
IHD	4.03 (2.40–6.77)	<0.001	0.6 (0.24–1.49)	0.27
AF	0.26 (0.11–0.62)	0.002	0.36 (0.14–0.93)	0.03
Valvular HD	0.76 (0.30–1.93)	0.56		
Indication		<0.001		0.004
Sinus node dysfunction	1 (ref)		1 (ref)	
AV block	0.33 (0.13–0.85)		0.19 (0.07–0.49)	
Primary prevention for VT/VF	2.91 (1.64–5.16)		1.35 (0.52–3.51)	
Secondary prevention for VT/VF	0.15 (0.02–1.16)		0.22 (0.03–1.78)	
Cardiac resynchronization	0.57 (0.13–2.54)		1.38 (0.29–6.62)

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
