# Peer review of "Arrhythmias after SARS-CoV-2 Vaccination in Patients with a Cardiac Implantable Electronic Device: A Multicenter Study"

_biomedicines, 2022, doi:10.3390/biomedicines10112838_

Round 1
Reviewer 1 Report
Dear Authors,
I have read your interesting work entitled “ Arrhythmias after SARS-CoV-2 Vaccine vaccination in patients with cardiac implantable electronic device: a multicenter study”. Please find my comments below.
Line 62-63 – I propose adding basic address details to the name of the hospitals
Line 66 – The IRB abbreviation is not explained
Line 84 – explanation for AICD is needed
IRR and aIRR shortcuts are not explained
Paragraph 2.4 – please provide precise definitions of sustained and not sustained VA. Were there any differences in the definition of VA depending on the type of device and manufacturer? How were VF episodes treated by the device – shock or anti-tachycardia pacing? From the clinical point of view, there is a significant difference between treated VF episodes and ventricular high-rate episodes due to fast supraventricular rhythm.
A description of SVT should be provided similarly. Patients with atrial fibrillation should be precisely described. Were in this group (with FA) pts with continuous arrhythmia or paroxysmal form only? What about ventricular arrhythmia in pts with FA?
I think PMT episodes should be excluded from the analysis (or analyzed separately) due to another reason for the arrhythmia compared to supraventricular or ventricular tachycardia. If you want to include PMT in the analysis, please justify it.
Please provide precise characteristics of devices: numbers of peacemakers/ ICD/CRT – singe/dual chamber, and manufacturers - different manufacturers define and detect arrhythmia episodes differently.
I did not find data regarding sex; how many males/females were in the study group?
How many were ventricular arrhythmia episodes observed in PMT vs ICD/CRT population?
Even assuming that the purpose of the work was the incidence of arrhythmia per se, there are different populations regarding patients with implanted pacemakers vs CRT/ICD. I suggest adding data regarding LVEF and sex. How many females have decreased LVEF? Maybe lower VA incidence in females was related to higher LVEF and a lower rate of implanted ICDs?
Please try redesigning Table 3; it is currently not understood and is illegible. Please standardize the notation of the numbers (e.g. 3 or 3.0)
Interpretation of a lower incidence of VA in CKD can only be an assumption, as there is no data regarding differences in the treatment.
How was myocarditis ruled out in the study group? There is a large paragraph in the discussion regarding myocarditis, concluding that there was no myocarditis in the study group. However, the methodology did not specify how myocarditis was ruled out.
Author Response
Thank you for the thoughtful input and review of our manuscript. The manuscript was revised based on your suggestions and attached is our point-by-point response.
Thank you for your time and consideration. We are looking forward to hearing from you.
With many thanks for your attention, I remain.

Reviewer 2 Report
Dear Authors, congratulation, your manuscript is very interesting! It is a multicenter study on the occurrence of arrhythmias in patients with CIED after the COVID-19 vaccine. Post-vaccination period was found to be a factor associated with SVT. The manuscript is well written and very interesting, however, I have a few comments:
- In the abstract, when mentioning SVT for the first time, it should be written in the entire form, supraventricular tachycardia.
- In the discussion, it should be stated that the data provided are not a demonstration of a causal relationship between vaccine and cardiac arrhythmias.
- Do the Authors think that the inflammation following the vaccination could be an explanation for the association between vaccine and SVT? Are there any other possible pathophysiological mechanism?
- The reference list should be implemented with the following: 10.3390/vaccines10020308; 10.1016/j.jjcc.2021.11.019; 10.3390/DIAGNOSTICS11091647; 10.1055/s-0042-1742970
Author Response

(The authors gave the same response as above.)

Round 2
Reviewer 1 Report
Dear Authors,
I have read the corrected version of the manuscript. I think that current version is sufficiently improved.
Sincerely yours
Author Response
Thank you so much for your thoughtful input. We believe our manuscript will provide insightful information to our readers.